# Marine Bacteria versus Microalgae: Who Is the Best for Biotechnological Production of Bioactive Compounds with Antioxidant Properties and Other Biological Applications?

**DOI:** 10.3390/md18010028

**Published:** 2019-12-29

**Authors:** Masoud Hamidi, Pouya Safarzadeh Kozani, Pooria Safarzadeh Kozani, Guillaume Pierre, Philippe Michaud, Cédric Delattre

**Affiliations:** 1Food and Drug Research Center, Vice-Chancellery of Food and Drug, Guilan University of Medical Sciences, Rasht P.O. Box 41446/66949, Iran; m.hamidi2008@gmail.com; 2Department of Medical Biotechnology, Faculty of Paramedicine, Guilan University of Medical Sciences, Rasht P.O. Box 44771/66595, Iran; puyasafarzadeh@gmail.com; 3Department of Medical Biotechnology, Faculty of Medical Sciences, Tarbiat Modares University, Tehran P.O. Box 14115/111, Iran; pooriasafarzadeh@gmail.com; 4Université Clermont Auvergne, CNRS, SIGMA Clermont, Institut Pascal, F-63000 Clermont-Ferrand, France; guillaume.pierre@uca.fr (G.P.); philippe.michaud@uca.fr (P.M.); 5Institut Universitaire de France (IUF), 1 rue Descartes, 75005 Paris, France

**Keywords:** antioxidant, bacteria, carotenoids, exopolysaccharides, microalgae

## Abstract

Natural bioactive compounds with antioxidant activity play remarkable roles in the prevention of reactive oxygen species (ROS) formation. ROS, which are formed by different pathways, have various pathological influences such as DNA damage, carcinogenesis, and cellular degeneration. Incremental demands have prompted the search for newer and alternative resources of natural bioactive compounds with antioxidant properties. The marine environment encompasses almost three-quarters of our planet and is home to many eukaryotic and prokaryotic microorganisms. Because of extreme physical and chemical conditions, the marine environment is a rich source of chemical and biological diversity, and marine microorganisms have high potential as a source of commercially interesting compounds with various pharmaceutical, nutraceutical, and cosmeceutical applications. Bacteria and microalgae are the most important producers of valuable molecules including antioxidant enzymes (such as superoxide dismutase and catalase) and antioxidant substances (such as carotenoids, exopolysaccharides, and bioactive peptides) with various valuable biological properties and applications. Here, we review the current knowledge of these bioactive compounds while highlighting their antioxidant properties, production yield, health-related benefits, and potential applications in various biological and industrial fields.

## 1. Introduction

Antioxidant properties have become a very crucial subject over the past two decades and also the topic of exhaustive research all over the world because of food and pharmaceutical industry demands, which have investigated natural antioxidant sources to produce efficient antioxidant biomolecules as anti-carcinogenic and anti-aging agents [1,2,3,4]. It is well-described that during oxidative stress, specific molecules called reactive oxygen species (ROS), such as hydroxyl radical, hydrogen peroxide, nitric acid, or superoxide anions, are generated [1,2,3]. Consequently, during cellular metabolism, some cellular tissue damage can be observed due to the ROS effects on biological macromolecules such as lipids, proteins, and nucleic acids [1,2,3,4]. It has been reported that these ROS play significant roles in carcinogenesis and diverse health disorders such as (i) lung injury, (ii) aging, (iii) diabetes mellitus, (iv) inflammatory diseases, and (v) atherosclerosis [1,2,5,6,7,8,9,10,11,12].

From an evolutionary point of view, all living organisms have developed enzymatic and non-enzymatic defense systems, such as the synthesis of (i) glutathione peroxidase/reductase, (ii) superoxide dismutase, (iii) vitamin C and E, (iv) thioredoxin, and (v) glutathione, to diminish the degree of damage caused by ROS [1,2,3,6]. Nevertheless, these innate antioxidant biomolecules are not abundant enough to completely prevent oxidative damage. Antioxidant molecule additives are generally used to fight against oxidative damage and to protect human cells by regulating the quantity of ROS. The food industry generally utilizes chemical antioxidants such as butylated hydroxytoluene (BHT), tert-butylhydroquinone (TBHQ), propyl gallate (PG), or butylated hydroxyanisole (BHA) [6,7,8,9,10,11,12,13]. Nonetheless, their use in food and their safety profile are increasingly controversial due to causing liver damage and their potential impact in carcinogenesis [6,7,8,9,10,11,12,13,14,15]. This is why, during recent years, safe and natural antioxidant molecules extracted from bioresources are becoming more advantageous in substituting the use of chemical and synthetic antioxidant agents in different fields.

For a long time, most of the antioxidant ingredients such as phenolic compounds, pigments, polysaccharides, and carotenoids were extracted from plants, fruits, and seaweeds [16,17,18,19,20,21,22]. Concerning seaweed, relative polysaccharides and pigment compounds have been validated as efficient antioxidants through various techniques such as 1,1-diphenyl-2-picrylhydrazyl radical (DPPH•) scavenging assay, hydroxyl/superoxide radicals (•OH/O_2_^−^•) scavenging assay, ferric reducing antioxidant power (FRAP), or lipid peroxidation inhibition capacity assay [18,21,22,23]. 

In the immensity of marine biotopes, a large amount of new efficient natural antioxidants continue to be discovered. In fact, the marine environment covers at least three-quarters of the earth’s surface, and many microorganisms inhabiting it under extreme conditions have developed very intelligent strategies to grow and adapt. Therefore, marine organisms constitute an inexhaustible source for antioxidant biomolecules among which we can mention (i) enzymes (catalase and superoxide dismutase), (ii) exopolysaccharides, (iii) carotenoids, and (iv) peptides [24,25,26,27,28,29]. The constant growth of international markets, asking for more natural antioxidant biomolecules, has to be correlated with the industrial expansion of marine microorganisms for high scale-up industrial processes to produce better yields of selected marine antioxidants.

Consequently, this present review aims to give an overview of the advances in the field of antioxidant biomolecules derived from marine microorganisms such as bacteria and microalgae, as well as their current development for potential industrial applications in nutraceutical, cosmeceutical, and pharmaceutical fields.

## 2. Description of Marine Organisms

### 2.1. Marine Bacteria

Bacteria were one of the first forms of life that appeared on earth, and to this day they can be found almost everywhere, from land to marine environments such as oceans and seas. They comprise a wide range of prokaryote microorganisms. Bacteria are very small unicellular organisms (a few micrometers in length) and they appear in various shapes and forms such as spheres, rods, and spirals, and they can live in different environments with various and sometimes extreme conditions such as high temperatures or high salinity. They can also live symbiotically and parasitically with plants and animals, and are crucial components of different ecosystems. Estimations suggest that the earth happens to contain 10^9^ taxa of bacteria, whereas the oceans contain 10^6^ [30]. As the most widespread and varied members of the microbial groups in oceans, bacteria have important roles in biogeochemical pathways and fluidity of energy and matter [31]. Marine environment-isolated bacteria are believed to be dissimilar from their earth-living counterparts in physiological, biochemical, and molecular properties and, consequently, they may produce different metabolites [32]. Marine bacteria inhabit surface waters, from coastal to offshore areas, including general oceanic areas such as blue waters and special condition areas such as hot thermal vents. They have a distribution of 10^5^ cells per milliliter (mL), and it is calculated that the oceans altogether contain 3.6 × 10^29^ microbial cells, and that most of this biomass is composed of bacteria, archaea, and other microorganisms such as protists and unicellular fungi. *Pseudomonas* sp., *Vibrio* sp., *Achromobacter* sp., *Flavobacterium* sp., and *Micrococcus* sp. are known as the principal seawater bacteria [30]. 

As reports clarify, of all bioactive metabolites that microorganisms produce, about 40–45% are generated by Actinomycete bacteria (soil-derived genera, *Streptomyces*, and *Micromonospora*) and are currently utilized in clinical fields [32]. Over 50% of microbial antibiotics are comprised of metabolites derived from these bacteria [33]. On the other hand, not so much is understood about the capabilities of the compounds derived from deep-ocean bacteria [33]. The speed of discovering compounds from old microbial drug producers, such as actinomycetes and hyphomycetes, is declining, and now it is time to reach out to marine bacteria and use their capacities [33]. Research conducted in the past has reported different varieties among the seaweed epiphytic bacterial flora. Lately, scientific communities have paid attention to these bacteria because of their potential for being a source of secondary metabolites with marketing importance [34,35]. *Bifurcaria bifurcata* epiphytic bacteria are an exceptional source of natural antioxidant and antimicrobial compounds [34]. Some bacteria are believed to have distinct physiological and biological roles due to the pigments they contain [35]. Several newly conducted research studies on pigments, such as violacein, astaxanthin, canthaxanthin, zeaxanthin, rubrolone, and carotenoid derivatives of bacteria from marine sources, confirm their effective radical scavenging activity [35]. Marine organisms living in environments with extreme conditions such as high salinity, low temperature, or extreme pressure have developed specific metabolites to overcome factors threatening their survival, proliferation, facilitated storage, transportation, and turnover of key biological elements [36]. Several pieces of research have reported that the carotenoids produced by halophilic bacteria [37,38], mostly bacterioruberin, show anti-cancer and antioxidant activities [39,40,41,42,43,44,45].

### 2.2. Microalgae

As unicellular and photosynthetic microorganisms, microalgae usually live in aqueous environments such as freshwater, blackish, and marine systems, and also non-aqueous environments such as soil and air. They live in both sediments and water column, and some of them are airborne microalgae or photosymbionts with other marine organisms for example mollusks, corals, or sponges [46,47,48]. As photoautotrophic microorganisms, microalgae use light and inorganic nutrients such as phosphorus, nitrogen, and CO_2_ to synthesize complex organic macromolecules such as pigments, lipids, nucleic acids, polysaccharides, and proteins, and almost half of the atmospheric oxygen is produced by microalgae [49]. Microalgae are mostly categorized into two groups: prokaryotes such as Cyanobacteria and Prochlorophyta, and eukaryotes such as Chlorophyta, Rhodophyta, Phaeophyta, Bacillariophyta, and Chrysophyta [50]. Estimations indicate that there are about 800,000 species of eukaryotic microalgae and cyanobacteria, of which only 50,000 species have been identified so far [51,52,53]. Chrysophyta (golden algae), Chlorophyta (green algae), Cyanophyta (blue-green algae), and Bacillariophyta (diatoms) are the four most important categories of microalgae [54]. Microalgae can grow in freshwater, marine, and highly saline environments [55]. Many microalgae species have short reproduction periods, so they do not have to compete for food production, and they can be simply cultivated in significant amounts in closed bioreactors, guaranteeing a theoretically limitless supply of biomass [50,56]. Additionally, using clean nutrient media for microalgae culturing grants the ability to control the quality of the microalgal cells, keeping them free of herbicides, pesticides, or any other toxic substances. On the other hand, microalgae could be used to take up valuable nutrients from wastewater [56]. Microalgae culture offers a noteworthy step for wastewater treatment because they provide a tertiary biotreatment coupled with the production of possibly valuable biomass. In fact, using microalgae for wastewater treatment emerges from their different properties such as fast growth and nutrient removal capacity. They have this ability to use inorganic nitrogen and phosphorus for their growth. Microalgae can be utilized in wastewater treatment for several different aims, such as reduction of biochemical oxygen demand (BOD), removal of nitrogen and/or phosphorus, inhibition of coliforms, and removal of heavy metals and other toxic organic compounds [57]. 

The relative simplicity of desired compound purification has accentuated the importance of microalgae as a natural source of bioactive products [58]. A unique property of microalgae is their chemical composition differences dependent on several environmental factors such as temperature, salinity, illumination, pH value, mineral content, CO_2_ supply, population density, growth phase, and physiological status; therefore, maximum production of a particular group of chemical compounds could be feasible by optimizing these environmental growth conditions [59]. It is important to state that under normal conditions, microalgae will not produce biologically valuable metabolites or the production yield will become very low. On the other hand, the biosynthesis of these compounds is inducible by exposing microalgae to stressful conditions such as high salinity, strong light, nitrogen deprivation, high temperature, short-term UV radiation, or a mixture of these different conditions. Actually, this property of microalgae is believed to be a very essential element in improving the production of carbohydrates, lipids, carotenoids, and other products [60].

Cyanobacteria are organisms that have unique abilities and are used as food, animal feed (50% of the present-day world *Arthrospira* production), and fuel [61,62,63]. In the old days, cyanobacteria were famous as blue-green algae because of their phycocyanin pigments, which are specific accessory photosynthetic pigments alongside chlorophyll a [63]. Cyanobacteria are a well-known potential and important class of organisms for the extraction of unique, new, and biochemically active natural products. Cyanobacteria such as *Spirulina*, *Anabaena*, *Nostoc*, and *Oscillatoria* produce great amounts of secondary metabolites that contain 40% lipopeptides, 9% amides, 5.6% amino acids, 4.2% fatty acids, and 4.2% macrolides [58]. Lipopeptides of cyanobacteria have various properties such as cytotoxic (41%), antitumor (13%), antibiotic (12%), and antiviral activities (4%), and the activity of the remaining 18% is considered as antimalarial, antimycotics, multi-drug resistance reverser, antifeedant, and immunosuppressive [58]. 

*Spirulina* is a cyanobacterium that grows in waters with alkaline conditions and has been used as food in Africa for many centuries because of its nutritional benefits, whereas nowadays it is globally utilized as a nutraceutical food supplement. In recent years, many pieces of research have worked on estimating the positive therapeutic effects of *Spirulina* on a wide range of diseases such as hyperglycerolemia, hypercholesterolemia, cardiovascular diseases, inflammatory diseases, cancer, and viral infections. In fact, hypolipidemic, antioxidant, and anti-inflammatory activities of *Spirulina* contribute to its anti-cardiovascular effects. The long history of *Spirulina* consumption as a food source and its approved safety in animal studies backs up the idea of its safety for human applications [64]. Among many *Spirulina* species, three species including *Spirulina platensis* (*Arthrospira platensis*), *Spirulina maxima* (*Arthrospira maxima*), and *Spirulina fusiformis* (*Arthrospira fusiformis*) are highly investigated because they are edible and have nutritional and therapeutic benefits [65,66,67,68].

From thousands of years ago, microalgae have been utilized for nutritional purposes; as an example, *Nostoc* was consumed to overcome famine in China, Chad, and Mexico. Even though microalgae have been used in traditional ways, microalgae culture is still a modern-day technology [55].

From the beginning of the century, the market size of functional foods derived from microalgae has experienced a five-fold rise and its growth has now relatively matured [41]. Industrial large-scale culture commenced in Japan in the early 1960s with the culture of *Chlorella* by Nihon Chlorella (Taipei, Taiwan) [69]. Nowadays, microalgae are consumed both as dried whole algae and also as the source for the extraction of worthwhile food supplements such as omega-3 fatty acids and carotenoids [41]. Microalgae products with high nutritional values are available in two forms: one in pure form as extracts, tablets, or capsules, and the other as additives to several food products, such as gums, candy bars, beverages, and pasta [60]. Besides their utilization in human nutrition, microalgae can be added into the feed of a variety of animals from fish to pets, as well as farm animals. About 30% of the present-day world microalgal production is utilized for animal feed applications [61,62]. So far, more than 15,000 new compounds, derived from algal biomass, have been chemically clarified [70]. Commonly, microalgae contain 40–70% proteins; 12–30% carbohydrates; 4–20% lipids; 8–14% carotene; and considerable amounts of vitamins B1, B2, B3, B6, B12, E, K, and D [59,71]. Production of potentially marketable compounds such as β-carotene; astaxanthin; polyunsaturated fatty acids (PUFA) such as docosahexaenoic acid (DHA) and eicosapentaenoic acid (EPA); and polysaccharides such as β-glucan, sterols, chlorophylls, and phycobiliproteins account for the therapeutic and nutritional supplements derived from microalgae [41,72]. Various properties of an alga species such as size, shape, toxicity, and digestibility determine its nutritional value [70].

Antioxidant molecules have gained great interest lately [41], and microalgae stand for a nearly intact resource of natural antioxidants because of their vast biodiversity, which is much more diverse than higher plants [73]. On the other hand, not all groups of microalgae can be applied as natural sources of antioxidants because of their broadly diverse content of products and non-favorable growth rate, as well as some other factors [55]. The leading genera of microalgae in industrial production are *Chlorella*, *Isochrysis*, *Chaetoceros*, *Dunaliella*, *Haematococcus,* and *Schizochytrium* [69,74]. Additionally, prokaryotic microalgae such as *Arthrospira* and *Spirulina* and eukaryotic microalgae such as *Chlorella* are the most commercially valuable microalgae in the field of health food and nutrition supplements [41,75]. There are several reports regarding the antioxidant activity calculations corresponding to various kinds of microalgae. For instance, *Scenedesmus obliquus* strain M2-1 shows the highest total antioxidant capacity (149 ± 47 antioxidant activity unit) of intracellular extracts [76]. According to Guedes et al. who examined the antioxidant capacity of 23 microalgae species, *Scenedesmus obliquus* strain M2-1 was the one with the highest total antioxidant activity [76]. Furthermore, specific interest is enforced to isolate microalgae from extreme-condition environments such as hot springs and apply them as good sources of natural products for various biotechnological aims [77,78,79].

#### 2.2.1. Chlorella 

*Chlorella* is a kind of microalgae generally discovered in freshwater environments. It is a reliable source of antioxidants and natural bioactive compounds utilized in the food and pharmaceutical market [80]. They are unicellular photosynthetic microalgae having chlorophyll in their chloroplasts and they also contain lutein and other primary carotenoids such as α-carotene and β-carotene [81]. The most significant substance in *Chlorella* is β-1,3-glucan, which is an active immune-stimulator capturing free radicals and decreasing blood lipids [55] with numerous other health-promoting properties such as positive effects on wounds, gastric ulcers, constipation, hypercholesterolemia, and atherosclerosis-prevention activity, as well as antitumor properties [62,82]. *Chlorella* can also be utilized as a food additive due to the flavor-regulating activity of its coloring agents [62,83].

#### 2.2.2. Dunaliella 

*Dunaliella salina* is utilized because of its β-carotene content that can consist of up to 14% of its dry weight [84]. Another broadly cultivated microalga is *Dunaliella tertiolecta*, which is greatly tolerant to salinity and is utilized for β-carotene harvesting, although it is also famous for the production of anti-carcinogenic compounds such as violaxanthin [55]. 

#### 2.2.3. Haematococcus 

*Haematococcus pluvialis* is a unicellular microalga living freshwater and distributed in many environments around the world [85]. The most progressed product of this microalga is astaxanthin, which is utilized as a food additive or dietary supplement [41]. Besides the fact that the synthetic form of astaxanthin holds more than 95% of the market, *Haematococcus*-produced astaxanthin can still be utilized as a natural food colorant [86].

## 3. Marine Antioxidant Biomolecules

### 3.1. Description and Generalities for Food and Pharmaceutical Applications

Because ROS-induced oxidative stress is the main reason of inflammatory events leading to many different medical conditions such as cancer, diabetes, neurodegenerative, and cardiovascular diseases, the pursuit for understanding the mechanisms of oxidative stress and also finding novel antioxidant compounds have become the interest for medical scholars [50,87]. Epidemiological investigations have proven a strong correlation between antioxidant utilization and a diminished risk of commonplace chronic diseases such as cardiovascular disease and cancer [88].

The principal concern of the biotechnological industry is to find antioxidant compounds from natural sources to replace artificial antioxidants such as BHA and BHT, which are used in medicinal preparations and packed foods [89,90]. The marine environment, covering nearly 70% of the earth’s surface, is a massive source of biodiverse resources including almost 300,000 known species, but it is estimated that this number is only a small percentage of the total number of species that are yet to be discovered. Luckily, investigations regarding marine microbial diversity and ecology, such as phylogenetic studies, have been achievable due to the development of culture-independent techniques such as small ribosomal RNA (rRNA) analysis and metagenomic approaches [30]. 

Marine microorganisms produce biologically active compounds with novel properties to adapt to extreme marine environmental conditions such as high or low temperature, alkaline or acidic water, and high pressure, as well as limited substrates of the deep-sea water [30]. So far, more than 25,000 of these compounds have been discovered [91], and from 1998 to 2006, around 592 of them have been reviewed for their antitumor and cytotoxic properties in preclinical and clinical studies [33].

Marine biotechnology aims to find efficient novel methods for the production of marine organism-related novel products. These products can have positive effects on human health and can be used as new medicines and antioxidants in the food industry, as well as biofuels in the energy industry [92].

Among different marine organisms, bacteria and microalgae represent themselves as important sources of bioactive compounds [91]. Consumption of microalgae for nutritional purposes is at least a thousand years old. Furthermore, microalgae culture development during recent decades allows for great amounts of algal biomass to be utilized for various applications [55]. Currently, global production of microalgae is predominately directed at various applications because algal biomass has great amounts of pigments, proteins, essential fatty acids, polysaccharides, vitamins, and minerals, which all have gained great attention in the production of natural products in both the nutraceutical and cosmeceutical industries [55]. Some microalgae species such as *Arthrospira* and *Chlorella* are well-known in the skincare market [90]. Face and skincare products are a field in which microalgal extracts are majorly utilized, including products such as antiaging creams, refreshing care products, and emollients [90]. Microalgal extracts are also used in sun protection and hair-care products [69].

Antioxidant properties of microalgal extracts result from their structural properties such as phytyl chain, a porphyrin ring, and conjugated double bonds [60]. Furthermore, the activity of each antioxidant varies in different oxidation systems (in vitro, ex vivo, and in vivo). Additional attempts are required to explain the connection between antioxidant activities and their physiological effects in biological systems [88]. The biological activities of antioxidants are tightly correlated with their absorption and metabolism. There are numerous complications in evaluating the true impacts of marine antioxidants. The main difficulties are the various kinds of antioxidants found in foods, their large differences in bioavailability, and the difficulties in finding an answer to the intricate absorption and metabolism mechanisms [88]. Even though some available data propose that health benefits can result from the use of marine natural products, additional epidemiological or clinical studies are nonetheless required to reinforce these statements with more confidence [33]. In fact, regardless of the developing market for antioxidants, uncertainty about the positive effects of dietary antioxidant consumption still lingers on. The relationship between the intake of antioxidant vitamins (for example, β-carotene, vitamin C, and vitamin E) or selenium supplements and the incidence and mortality of prostate cancer is still unclear on the basis of the results of a meta-analysis. Also, according to the meta-analysis of randomized controlled trials, antioxidant supplements do not avert skin cancer [33].

Table 1 gathers several famous carotenoids and their applications in different fields in terms of their benefits to human health, as well as some industrial applications.

### 3.2. Carotenoids

Carotenoids are lipophilic compounds appearing in different colors such as yellow, orange, and red. They are the most prevalent and frequent pigments found in nature [186,187] with more than 700 of them identified to this day [38,56]. Most carotenoids have a mutual C40 isoprene unit backbone structure, called terpenoid, and are categorized into two groups: carotenes and xanthophylls. Carotenes are only hydrocarbon and xanthophylls are the oxygenated byproducts of carotenes. Xanthophylls are quite hydrophilic compounds because of the hydroxyl and keto groups they have at the end rings [27]. The yellow, orange, and red hue coloring in higher plants, algae, bacteria, fungi, and some animals result from carotenoids. In all photosynthetic organisms, carotenoids are the principal non-chlorophyll accessory pigments that are necessary for light-harvesting and photoprotection [142]. Carotenoids are also capable of photo-protecting the photosynthetic machinery from extra light by capturing free radicals [188]. On the basis of direct or indirect engagement of carotenoids in photosynthesis, they are divided into two groups: primary carotenoids, such as α-carotene, β-carotene, and lutein, which are directly engaged in photosynthesis and are vital for cellular survival, and secondary carotenoids, such as astaxanthin and canthaxanthin, which are produced and stored in a process called carotenogenesis during the time that bacteria or microalgae are subjected to particular environmental stimuli [189,190].

A diverse range of commercial applications emerge from the biological properties of carotenoids [87]. Lately, significant attention has been drawn to carotenoids because of their biotechnological applications and their probable valuable applications in the field of human healthcare, food processing, pharmaceuticalsm and cosmetics [142,191]. They are also famous and accepted as valuable antioxidants with available proof that they are engaged in other biological functions such as gene expression regulatory effects on intracellular and intercellular signaling [192]. Furthermore, because of their coloring properties, they have customarily been used in food and animal feed. Additionally, they are also known to enhance consumer perception of quality, for instance, adding carotenoids to fish feed will pass on color to farmed salmon [87].

In a search performed in PubMed, Scopus, and Web of Science, where “carotenoids” and “human health” were used as keywords, about 6000, 3000, and 6000 results were respectively found, indicating the promising effects of carotenoids on human health [41]. It is important to state that these findings are still inadequate because only a fraction of them have been conducted in humans [41]. 

Marine bacteria and microalgae act as a massive source of diverse forms of carotenoids. In these microorganisms, carotenoids have different functions such as regulating mechanisms against oxidative stress, granting cellular colorings, and administering photoprotection. It is notable to emphasize that particular carotenoids such as salinixanthin or bacterioruberin are exclusively produced by specific extremophilic bacteria and microalgae [41]. Microalgae have lately been in the center of interest among other sources of carotenoids because of their many advantages, such as reasonably easy cultivation, rapid replicability, ecological sustainability, and capability of adaptation to changing environmental conditions by creating a wide range of secondary metabolites. Biosynthesis of carotenoids in both bacteria and microalgae could be induced by either modification of cultivation conditions or genetic engineering methods. 

Figure 1 and Figure 2 display simplified diagrams of several microalgae and marine bacteria and the carotenoids they prominently produce.

Green microalgae can produce all of the xanthophylls biosynthesized by higher plants alongside some additional ones. In cyanobacteria and oxygenic photosynthetic bacteria, most of the xanthophylls are related to chlorophyll-binding polypeptides of the photosynthetic apparatus [193]. It is also important to mention that carotenes and xanthophylls are produced inside the plastids of green microalgae and are stored exclusively there. On the other hand, cytoplasmic storage of secondary xanthophylls has also been detected in some green microalgae such as astaxanthin in *Haematococcus* sp., which suggests the existence of an extra-plastid site for carotenoid biosynthesis. Otherwise, exportation and storage of chloroplast-synthesized xanthophylls have been detected and explained by a proposed theory that they can be in reach of all cellular compartments that need them [194,195,196]. 

To this day, the in vitro synthesis of carotenoids has maintained their mass production. Nevertheless, the developing present-day standards for nature-loving routines and a healthy way of life have offered a search for natural bio-compounds as alternatives to synthetic ones. Recently, microalgae and bacteria biomass are being utilized as naturally accessible carotenoid sources. This is a great new way forward, in contrast to the in vitro synthesis of carotenoids, which is costly and produces compounds that are sometimes biologically inert [41]. Because the production of carotenoids might be generally simpler in marine bacteria and microalgae, they have been known as good natural sources for carotenoids among other regular natural sources [41]. The production of bioactive compounds from marine bacteria and microalgae has been recently studied using current methods such as transcriptomics, genomics, and proteomics [41]. Marine bacteria and microalgae, besides their ability to produce novel bioactive compounds, propose special potentials for carotenoid mass-production, for instance, the danger of contamination with different microorganisms is diminished because of the high-saltiness conditions utilized in their culture media. This characteristic is more highlighted when extreme halophilic marine microorganisms such as Haloarchaea are utilized as the common source of bioactive compounds [37,38,41]. 

Marine bacteria and microalgae that have the ability to produce carotenoids in high amounts have been the center of attention of researchers and biotechnology companies for the past 30 years. A reason for this attention is the high demand for the biocompounds that are derived from these natural sources, which in itself shows that consumers prefer natural products rather than synthetic forms [41]. The USA and Europe are the business markets where the interest in carotenoids is at its highest level [41], and the worldwide market for carotenoids is expected to reach as high as 1.7 billion United States dollars as of 2022 [41].

Astaxanthin and β-carotene, as the most prevalent carotenoids, are briefly discussed in the following sections.

Table 2 and Table 3 show several carotenoids produced by a few famous microalgae and marine bacteria, as well as the amount of their production evaluated in different studies under optimized conditions.

#### 3.2.1. Astaxanthin

Astaxanthin is a red-pink-colored xanthophyll carotenoid possessing two extra oxygenated gatherings on each ring, granting it improved antioxidant properties [152]. Astaxanthin has the capability to connect across the whole-cell membrane from within to the outside, which gives it a stronger biological activity than other antioxidants [41]. As reports claim, the antioxidant properties of astaxanthin are a few folds higher than those of other carotenoids and tocopherols. The valuable health impacts such as photoprotection of eyes and skin, anti-inflammatory characteristics, heart health improvement, and cancer prevention properties of astaxanthin may result from its strong antioxidant activity [88]. A few studies have shown that astaxanthin, when utilized as a dietary enhancer (E number E161j), has potential helpful consequences for human wellbeing. The biotechnological production of astaxanthin from different sources has been investigated in detail to accomplish its production on a vast scale for a few commercial applications. Astaxanthin products, in different dosage forms such as tablets, cases, syrups, oils, delicate gels, creams, biomass, and granulated powders, are widely utilized for commercial applications [41]. A commercial form of astaxanthin, named Zanthin, which is a supercritical carbon dioxide derived from *Haematococcus pluvialis*, has received U.S. Food and Drug Administration (FDA) approval for its use as a dietary supplement [234]. 

*Haematococcus pluvialis*, *Chlorella zofigiensis*, *Chlorella vulgaris,* and *Chlorococcum* sp. are known as the main natural sources of astaxanthin. The overall production yield attributed to *Haematococcus pluvialis* surpasses any other natural sources, accounting for up to 4–5% of its dry weight [235]. *Haematococcus pluvialis* is currently cultivated at industrial scales because it can produce more than 3 g of astaxanthin kg/L dry biomass [194]. It is also valuable to mention that astaxanthin, derived from marine sources, has higher levels of biological activities than those of its soil-derived counterparts [88]. 

Companies such as Cyanotech and Aquasearch have developed a two-stage strategy for producing astaxanthin by *Haematococcus* sp. The first stage, named the “green” stage, is performed under optimal growth conditions and produces green biomass. The second stage, named the “red” stage, subjects microalgae to unfavorable environmental conditions, leading to astaxanthin collection. In mass production facilities, this method produces 2.2 mg/L astaxanthin, whereas at the laboratory scale and under nonstop illumination, maximum astaxanthin production of 11.5 mg/L is achievable [236].

#### 3.2.2. β-Carotene

β-carotene is a strong orange-colored pigment mostly found in green-colored leafy plants (such as parsley, spinach, and broccoli), fruits (such as mandarin and peach), and a few vegetables (such as carrot and pumpkin) [237]. It is utilized as a food coloring agent (E number E160) [135] and also acts as a precursor and inactive form of vitamin A in nature (synthesized from carotenoids with the help of the enzyme β-carotene 15,150-monooxygenase). After biosynthesis, vitamin A conversion into retinoid takes place to prevent hypervitaminosis A [238,239]. 

Several β-carotene isomers such as *all*-*trans*-β-carotene as well as other minor carotenoids that have been extracted from *Dunaliella salina* (which is a halotolerant microalga) are known as good antioxidant molecules [234]. β-Carotene, which is the principal carotenoid produced by *Dunaliella salina* (more than 10% of its dry weight) [91], was the first algal product to be commercialized [234]. The optimum salinity of 22% (*w*/*v*), which is 7–8 times higher than seawater salinity, is suitable for the growth of *Dunaliella salina,* making it the most halophilic eukaryotic microalga to produce β-carotene and suggesting the potential of this microalga for open-air culture [234].

Investigations have demonstrated that besides antioxidant properties, β-carotene has many other benefits to human health, such as liver fibrosis prevention, provitamin A functionality, night blindness prevention, anti-neurodegenerative disease properties, and skin photoprotection against UV light, as well as acute and chronic coronary syndrome prevention.

#### 3.2.3. Other Carotenoids

In this section, we briefly discuss the carotenoids that are not widespread, as well as rare carotenoids that are only found in several specific extreme-condition marine bacteria and microalgae. 

So far, commercial production of some particular carotenoids from several specific microalgae has been reported—cases such as *Muriellopsis* sp. for lutein production because of its high growth rate and high lutein content (up to 35 mg/L) [194], as well as *Chlorella vulgaris* being reported as a high producer of lutein [208]. Also, *Chlorella ellipsoidea* is a good producer of violaxanthin alongside xanthophylls such as antheraxanthin and zeaxanthin [208]. 

Halophilic Archaea, from the families Halobacteriaceae and Haloferacaceae, have a red color that is mostly results from a special carotenoid they contain named “bacterioruberin”. As recent investigations regarding the production of bacterioruberin show, the production of this rare carotenoid might be simply improved by enforcing modifications on light frequency, oxygen availability, pH, salinity, or temperature [41,223]. Furthermore, Haloarchaea are also organisms that produce several rare carotenoids such as salinixanthin [37,38]. It is worth mentioning that some researchers such as Shindo and Gamone and their colleagues consider saproxanthin, sioxanthin, and siphonaxanthin as rare carotenoids in their works and articles [41].

New strains of marine bacteria from the family Flavobacteriaceae have been shown to have rare carotenoids such as saproxanthin and myxol with antioxidant and other novel properties. Saproxanthin or myxol might be effective in strengthening biological membranes, reducing permeability to oxygen, and improving protection against oxidation. The antioxidant activity of these two compounds is even higher than zeaxanthin and β-carotene, which are commonly used antioxidants. Nonetheless, these novel and rare carotenoids require meticulous assessments before their execution in cosmeceutical products [31]. 

It has been evident that many marine bacteria not only produce several acyclic C30 carotenoids with a 30-carbon skeleton, but that they also produce dicyclic or monocyclic C40 carotenoids [240,241]. New rare carotenoids with β-carotene skeleton have been extracted and identified from marine bacteria from the class α-proteobacteria, phylum Proteobacteria—carotenoids such as astaxanthin glucoside from *Paracoccus* sp. strain N81106 [242], 4-ketonostoxanthin 3′-sulfate from *Erythrobacter* sp. strain. PC6 [243], and 2-hydroxyastaxanthin from *Brevundimonas* sp. strain SD212 [244]. It is important to mention that these marine bacteria are also capable of producing astaxanthin [245].

### 3.3. Exopolysaccharides

#### 3.3.1. Exopolysaccharides from Bacteria

Exopolysaccharides (EPS) comprise a substantial part of dissolved carbon in the marine environment and are effectively engaged in biofilm growth. Different marine bacteria produce EPS based on their ecological niche and physiological demands, and these EPS have important roles such as (a) in adhesion and colonization to surfaces; (b) in protecting cells from extreme temperature, salinity, and osmotic pressure; and (c) in *quorum sensing* and biochemical interactions [246,247]. These mentioned roles and capabilities offer different advantages even beyond environmental conditions, and these marine bacteria EPS can be extracted, purified, and used in a variety of fields. One remarkable aim is the possible utilization of these ESPs to meet industrial demands [248]. 

In addition to the various properties reported in research, antioxidant activities have been described as a helping hand of research and industrial development for challenging areas such as cosmetics, pharmacology, and food-processing [31]. Importantly, to understand the renowned structure–function relationship between the reported functions and the associated EPS structural features, deep and detailed structural analysis must be performed. This is remarkably true because tremendously distinct bacterial EPS structures can exhibit “similar” antioxidant activities. In 2016, Wang et al. [249] drew attention to the basic mechanisms for in vitro antioxidant activity of EPS. Bacterial EPS have a chelation activity that is correlated with their antioxidant properties and is linked to their −OH, −SH, −COOH, −PO_3_H_2_, −C=O, −NR2, −S−, and −O− functional groups [250]. Particular side chains, such as 1→2, 1→4, or 1→6, can also control these properties in addition to rhamnose (Rha), fucose (Fuc), or mannan (Man) residues. Low molecular weight EPS exhibit better antioxidant properties due to a higher ratio of reducing terminals, which grants them the ability to better accept and/or eliminate free radicals [251]. We robustly suggest newcomers in the topic to read the comprehensive review of Wang et al. [249].

As we know, several marine bacteria have been reported for their production of antioxidant EPS [252], even if no noteworthy connection has been observed between these properties and the mud, deep-sea hydrothermal vents, or hypersaline marine environments they inhabit. As reported for *Pseudomonas stutzeri* 273, isolated from marine sediments, it has been recommended that anti-biofilm and anti-biofouling roles are correlated with antioxidant properties [253]. EPS 273, extracted from the aforementioned strain, is rich in glucosamine (GlcN), Rha, glucose (Glc), and Man at 35.4%, 28.6%, 27.2%, and 8.7%, respectively. Another EPS extracted from the marine mud bacterium *Bacillus licheniformis* exhibits antioxidant activities around 42% and 51% for O_2_^−^• and •OH scavenging, respectively [254]. In the investigation of bacterial antioxidant EPS isolated from Egyptian habitats, EPS from *B. licheniformis* was reported for its antioxidant properties, which were 97% against DPPH•. Six other EPS producing marine bacteria with antioxidant properties such as *Bacillus insolitus*, *Bacillus polymyxa*, *Bacillus marinus*, *Bacillus anthracis*, *Staphylococcus* sp., and *Bacillus brevis* were also discovered with 85, 84, 83, 81, 81, and 77% antioxidant activity against DPPH•, respectively [255]. In 2018, an EPS of 5.50 × 10^5^ Da from *Bacillus* sp. named MSHN2016, with considerable DPPH• scavenging activity, was extracted and characterized, which was composed of arabinose (Ara), xylose (Xyl), Glc, and glucuronic acid (GlcA) (1:1:2:1 relative ratio) [256]. Shivale et al. (2018) investigated the isolates from 156 EPS-producing bacteria from marine soil in the northwest coastal region of India [257], reporting that the EPS from *Bacillus brevis*, *Bacillus sorensis*, *Janibacter melonis*, and various species of Pseudomonas are an encouraging source of natural antioxidant metabolites. Again in India, an extracted EPS from the marine *Microbacterium aurantiacum* FSW-25 (named EPSMi-25) was found to have 80% DPPH• scavenging activity at 1 mg/mL. EPSMi-25 is an acidic sugar that exhibits rheological stability similar to xanthan, and is comprised of Glc, Man, Fuc, and GlcA with a molecular weight of 7 × 10^6^ Da [258]. 

Bacteria inhabiting deep-sea environments also produce EPS with antioxidant properties, for example, *Zunongwangia profunda* SM-A87, which is isolated from a place next to the southern Okinawa Trough at a water depth of 1245 m. SM-A87 EPS show DPPH•, •OH, and O_2_^−^• scavenging ratios of 48.5, 58.7, and 27.2%, respectively, for concentrations ranging from 0.1 to 10 mg/mL, which can result from their structure principally composed of GlcA and Fuc. The utilization of antioxidant EPS from polar bacteria is less reported, but some researchers have detected attractive structures, such as from the Arctic marine bacterium *Polaribacter* sp. SM1127 [259]. SM1127 EPS has a molecular weight of 220 kDa and is comprised of *N*-acetylglucosamine (GlcNAc), Man, GlcA, Fuc, Glc, and Rha residues, which are connected in a complex way. It exhibits encouraging scavenging ratios (10 mg/mL) for DPPH•, •OH, and O_2_^−^•, with values around 55.40%, 52.1%, and 28.2%, respectively. Also, *Edwardsiella tarda* has been reported to produce (1,3)-linked mannan with a molecular weight that ranges from 29 to 70 kDa [260].

Collectively, Table 4 gives a non-exhaustive overview of some marine bacteria that produce antioxidant EPS reported in the last decade.

#### 3.3.2. Exopolysaccharides from Microalgae

Polysaccharides, and more specifically soluble EPS, which are produced by several prokaryotic and eukaryotic microalgae, are at the center of attention of the academic and industrial communities due to their antioxidant and other novel properties [266,267]. Currently, highly valuable compounds, such as pigments and sulfated polysaccharides for industrial production, are only produced by 10 to 15 microalgae species, whereas other species are still being studied [49,266,267,268,269]. In fact, metabolism of aerobic organisms, as well as some exogenous agents, produce ROS such as superoxide radicals (O2•¯), singlet oxygen (^1^O_2_), hydroxyl radicals (•OH), and hydrogen peroxide (H_2_O_2_), which are believed to be the origin of some forms of cellular damage such as aging, mutagenesis, and carcinogenesis [270]. Photoautotrophic microalgae collect and store antioxidative scavenger complexes, such as polysaccharides, to protect their cells from long-time exposure to ROS and other free radicals [264]. Many polysaccharides that are derived from these microorganisms have been recognized as probable antioxidants (Table 5) [264,266,267,271]. Glycosidic linkages, monosaccharide composition, molecular weight, and conformation of antioxidant polysaccharides determine their scavenging activity [272]. The ability of ion chelation (uronic acids containing polysaccharides) and radical scavenging reducing power of polysaccharides are the two factors that define their antioxidant properties. It is also proposed that the antioxidant activities of some of these polysaccharides emerge through induction of gene expression of those which encode antioxidant enzymes such as superoxide dismutase or glutathione peroxidase. However, due to lack of available information on molecular mechanisms responsible for antioxidant activities, this property of polysaccharides remains debatable. Furthermore, some non-carbohydrate compounds with antioxidant properties, which are extracted alongside polysaccharides, are occasionally believed to be obstructions for the source of the detected scavenging activities [271]. The isolated antioxidant polysaccharides from prokaryotic microalgae such as *Spirulina platensis* and *Nostoc flagelliforme* and eukaryotic microalgae such as *Navicula* sp., *Rhodella reticulata*, *Porphyridium* sp., *Porphyridium cruentum, Dunaliella salina, Schizochytrium* sp., and *Isochrysis galbana* have effective scavenging activities on hydroxyl radicals, hydroxyl peroxide, and superoxide radicals [24,273,274,275,276,277,278]. The absence of appropriately defined structures is an obstacle on the way of polysaccharide application in the fields of food and health, even though in many studies the scavenging abilities of these polysaccharides have been verified. Additionally, commercial antioxidants such as BHA, propyl gallate, BHT, and tertiary butylhydroxyquinone, which are authorized as food antioxidants, show higher antioxidant activities than most of the microalgae-derived polysaccharides [279], a fact that restricts the potential application of the microalgae-derived polysaccharides. Several authors have suggested strategies such as culture condition regulations as a method for the enhancement of antioxidant activities of polysaccharide.

The Diatom *Navicula* sp. produces a sulfated EPS that is composed of Glc, Rha, Xyl, Man, and galactose (Gal) at several ratios on the basis of the wavelengths utilized to cultivate the strain [280], with polysaccharides derived from the white wavelength-irritated cultures having higher Rha and sulfated content, as well as a higher scavenging activity (DPPH•) in comparison to those derived from the red and blue wavelength-irritated cultures, resulting in higher molecular weights. Likewise, three polysaccharides that have been isolated from *Nostoc flagelliforme,* cultivated in normal, salt stress, and mixotrophic conditions, contain nine different monosaccharides with their uronic acid content, monosaccharide ratio, and molecular weights differing on the basis of the conditions of their culture. The polysaccharides isolated from mixotrophic culture conditions exhibit the highest antioxidant activity, even though all these polysaccharides demonstrate robust scavenging activity on hydroxyl and 2,2’-azino-bis(3-ethylbenzothiazoline-6-sulfonic acid) radicals (ABTS•) [273]. The thing that makes it challenging to accomplish an accepted structural adaptation of EPS to environmental conditions is the lack of complete characterization of polysaccharides to detect real structural differences. On the other hand, they pave the way for new strategies to optimize the antioxidant properties of microalgae-produced EPS. 

A mucilage made of partly soluble sulfated polysaccharides, which is sometimes non-covalently linked with glycoproteins, encapsulates the cell of red microalgae [24,267]. These polysaccharide layers increase in size on the basis of physicochemical environmental conditions and nutrient starvation, and can reach up to 50–60% of dry matter, which supports the idea that under stress conditions, these microalgae supply and utilize energy to synthesize exocellular sugars to survive. Some of the closely studied red microalgae are the strains that belong to the *Porphyridium* genus [267]. These species naturally inhabit shallow near-shore waters and sea sands, which are accompanied by stress conditions, such as salts, temperature, and light, resulting in ROS formation. Tannin-Spitz et al. (2005) have proven the protective role of the EPS from *Porphyridium* sp. against oxidation by showing their capability to restrain in vitro autoxidation of linoleic acid, ferrous oxidation, and the oxidative damage to 3T3 cells caused by FeSO_4_ [24]. Furthermore, Sun et al. (2009) verified the scavenging abilities of the polysaccharides from *Porphyridium* species on free radicals and their inhibitory effects on ascorbic acid- and FeSO_4_-induced lipid peroxidation in liver homogenates and mouse erythrocyte hemolysis using low molecular weight forms obtained after microwave degradation [96]. It is important to state that these authors did not find substantial antioxidant abilities with the crude high molecular weight polysaccharide fractions.

### 3.4. Others Bioactive Compounds from Marine Bacteria and Microalgae

#### 3.4.1. Bioactive Peptides

Recently, several bioactive peptides have been detected in a variety of marine protein resources that possess antioxidant properties. Because marine bacteria and microalgae are full of proteins but have low amounts of fat, they are known as reliable sources of new antioxidant peptides [283]. Many reports have declared that antioxidant peptides usually have 3–20 amino acid residues [284] and have positive effects on human health, and thus could be used in the food industry. Antioxidant activity of bioactive peptides can result from their free radical scavenging activity, metal ion chelating capability, and lipid peroxidation inhibition [283]. Because of the sequential, structural, and compositional properties of these peptides, besides antioxidant activities [285,286], they might show various kinds of bioactivities such as antihypertensive [287], immunomodulatory effects [288,289], and anticancer and antimicrobial activity [283]. Peptides extracted from marine bacteria and microalgae could be used as food additives to improve consumer health and shelf life of food products [290].

Recently, some algae proteins such as *Chlorella vulgaris*, *Chlorella ellipsoidea*, and red algae *Palmaria palmata* waste proteins have been discovered to be close relative proteins of antioxidant peptides [283]. For example, the peptide Valine-Glutamic acid-Cysteine-Tyrosine-Glycine-Proline-Asparagine-Arginine-Proline-Glutamine-Phenylalanine, released from pepsin hydrolysate of *Chlorella vulgaris*, exhibits the greatest ABTS• scavenging activity in comparison to Trolox, ascorbic acid, and BHT [291]. 

According to investigations regarding the antioxidant activity of foods, it has been found that enzymatic digestion of different marine food proteins, such as *Spirulina platensis*, *Dunaliella salina*, *Botryococcus braunii*, dried bonito, dried salted fish, fish sauce, and fish water-soluble proteins results in the production of several antioxidant peptides [292]. Many human and animal studies have revealed that *Chlorella* intake gives rise to robust antioxidant potential, possibly resulting from their active peptide content, but we have to state that the mechanism is still undetermined [293]. Hydrolysates and peptides with antioxidant activities from marine bacteria and microalgae can be consumed as food additives to diminish oxidative changes occurring during storage in different food products. They also havve exclusive techno-functional properties, such as low viscosity, high solubility, and resistance to gel formation [290].

#### 3.4.2. Mycosporine-Like Amino Acid and Scytonemin 

Mycosporine-like amino acids (MAAs) are a family of secondary metabolites with a molecular weight of less than 400 Da. MAAs are colorless, uncharged, and water-soluble ampholytes with similar chemical structures differing in amino acid composition and are produced by bacteria and microalgae living in high sunlight habitats as a response to solar radiation [294]. Both prokaryotic and eukaryotic microalgae can be a source for the extraction of MAAs with the capability to be utilized in sunscreens. Additionally, mycosporine-glycine has great antioxidant, anti-inflammatory, and antiaging activities, as previous research disclosed, which offers a new understanding for the use of MAAs in cosmeceutical fields [31]. 

Furthermore, other photoprotective compounds such as Scytonemin, which is a dimer of indolic and phenolic subunits capable of decreasing the risk of damages resulted from UV light, have been isolated from prokaryotic microalgae [55].

#### 3.4.3. Vitamins, Mannosylglycerate, Phycoerythrobilin, and Phycobiliproteins

Molecules such as polyphenols, fatty acids, tocopherol, flavonoids, alkaloids, polyketides, glycosides, isoprenoids, catalases, and superoxide dismutase, which play key roles in the control of oxidative processes, could be derived from marine bacteria and microalgae [31,74,235]. Vitamin E is believed to be an applicable antioxidant and is commonly used in the formulation of cosmetic products. Because microalgae, such as *Dunaliella tertiolecta* and *Tetraselmis suecica*, produce great amounts of α-tocopherol and vitamins, they could be valuable vitamin E sources [55].

Hyperthermophilic bacteria produce mannosylglycerate (MG), which is a biocompatible solute. In yeast models, MG has inhibitory effects on α-synuclein aggregates in Parkinson’s disease (PD). It is also considered as a good candidate for the treatment of PD because it can induce proper folding of α-synuclein and prevent its pathological aggregation [295]. 

Phycoerythrobilin, which is a red phycobilin isolated from *Spirulina* and *Porphyridium*, exhibits antioxidant properties, and it also could be applied in the decorative cosmetics industry for making products such as eye-liners and lipsticks. Phycobiliproteins isolated from *Porphyridium aerogineum*, which are light energy capturing protein complexes covalently bound to phycobilins, are also utilized in the food and cosmetic fields as colorants without showing any alterations due to pH fluctuations (4 to 5) and having color stability under light [55]. 

#### 3.4.4. Ovothiols

Ovothiols are sulfur-containing natural compounds biosynthesized by many marine organisms and microorganisms including microalgae such as *Dunaliella salina* and *Euglena gracilis*, as well as many marine bacteria such as *Shewanella*, *Photobacteria*, and *Pseudoalteromonas*. These compounds own unique chemical properties and various cellular functions. They show protective activity against oxidative stress, act as molecular messengers for inter- and intracellular signaling pathways, and serve as building blocks of more complex structures [296,297,298].

They enable organisms to survive changing environmental conditions and play a crucial role in maintaining cellular redox homeostasis [299,300]. 5-Thiohistidines act as outstanding nucleophiles and reducing agents, providing cellular protection against reactive oxygen and nitrogen species, electrophilic organic compounds, and thiophilic metals [296].

Ovothiol A (5(Nπ)-methyl thiohistidine) is one of the most abundant marine sulfur compounds. Due to the unique antioxidant properties [301,302,303] and broad distribution among marine microorganisms such as microalgae and Proteobacteria, it is believed that ovothiol A and its derivatives, ovothiol B and C, play important roles in cellular biochemistry [304,305]. 

Studies have demonstrated that some ovothiol derivatives act as neuroprotective agents for the mammalian brain. Furthermore, ovothiol A is believed to be an antioxidant due to its ROS and peroxynitrite scavenging activity and the ability to produce nitric oxide that can modulate endothelium functionality. Therefore, ovothiols may have anti-atherogenic properties and therapeutic potential in pathologies related to cardiovascular diseases caused by oxidative/inflammatory stress and endothelial dysfunction [296].

## 4. Conclusions

So far, among several biosources of marine bioactive compounds (carotenoids and ESP) with antioxidant properties and other biological applications, microalgae have been in the center of interest because of their rapid replicability, ecological sustainability, simplicity of cultivation, and capability of adaptation to changing environmental conditions. It is also worth mentioning that, in many cases, the carotenoid and EPS production yield of microalgae is much higher than marine bacteria, which makes them a more cost-effective option for the biotechnological scale production of the mentioned bioactive compounds. Additionally, modification of cultivation conditions and genetic engineering methods could also play key roles in increasing the production yield of both marine bacteria and microalgae. On the other hand, one other thing that must always be considered is the antioxidant activity and other properties of these bioactive compounds. Even though they might have the same structural, compositional, and sequential properties, microalgae-derived bioactive compounds show a higher bioactivity rate as compared with their marine bacteria-derived counterparts. Furthermore, for other rare bioactive compounds (other than carotenoids and ESP), it is hard and almost impossible to decide between marine bacteria and microalgae as a source for the biotechnological scale production, and because for production of each specific compound, a different type of microorganism might be suitable, we purpose that further investigations need to be conducted to determine the best producer for a specific type of bioactive compound.

## Figures and Tables

**Figure 1 marinedrugs-18-00028-f001:**
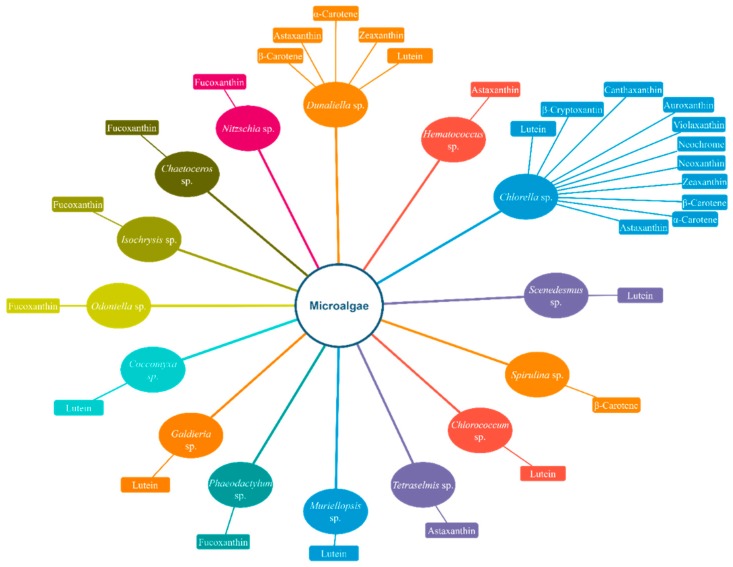
Several famous microalgae and the carotenoids produced by them. Only carotenoids produced at a high amount are mentioned in the diagram.

**Figure 2 marinedrugs-18-00028-f002:**
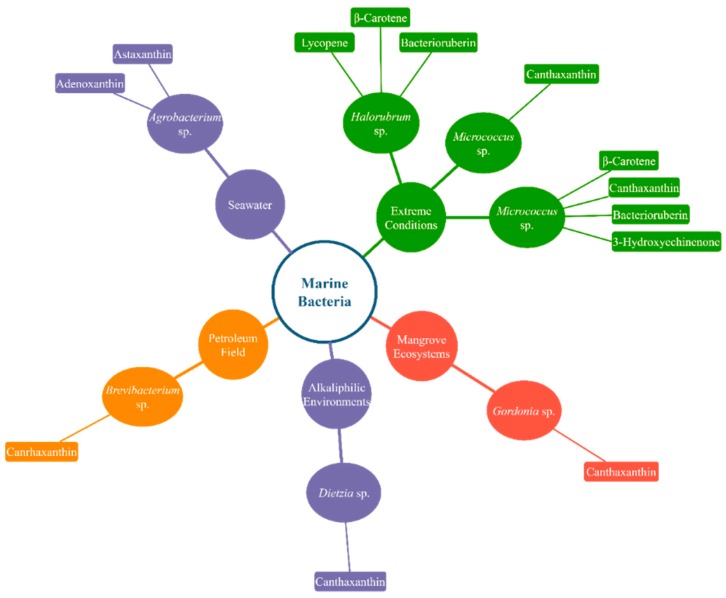
Various carotenoid-producing marine bacteria and their natural growth condition. Only a few well-known marine bacteria and the carotenoids they produce at a high rate are featured in the diagram.

**Table 1 marinedrugs-18-00028-t001:** Several carotenoids and their applications in different fields.

Carotenoids	Examples of Biological Properties, Functions, and Benefits to Human Health	Other Applications	Reference
Astaxanthin	Antioxidant properties Anti-cancer Anti-male infertility Anti-hepatotoxicity effects Anti-inflammatory effects Anti-diabetic nephropathy Anti-cardiovascular diseases Anti-neurodegenerative diseases	As a food coloring agent As feed additives	[41,93,94,95,96,97,98,99,100,101,102,103,104,105,106,107]
Lutein	Antioxidant properties Anti-cancer Stroke prevention Ocular protective effects Retinitis prevention Anti-diabetic retinopathy Anti-cardiovascular diseases *Helicobacter pylori* gastric infection prevention Cataract and age-related macular degeneration prevention	As a food coloring agent Used in poultry feeding	[41,108,109,110,111,112,113,114,115,116,117,118,119,120,121,122,123]
β-carotene	Antioxidant properties Liver fibrosis prevention Provitamin A functionality Night blindness prevention Anti-neurodegenerative diseases Skin photoprotection against UV light Acute and chronic coronary syndromes prevention	As a food coloring agent As food supplements	[98,106,124,125,126,127,128,129,130,131,132,133,134,135,136]
Lycopene	Antioxidant properties Anti-cancer Gene regulation activity Antiulcer activity Immune modulation Radiation protection	As a food coloring agent	[39,41,137,138,139,140,141]
Canthaxanthin	Antioxidant properties Anti-cardiovascular diseases Tan color creation Antitumoral activity Provitamin a functionality Immune system stimulation	As a food coloring agent As an additive for the feed of dogs, cats, ornamental fish, birds, and other pets	[39,41,81,107,142,143,144,145,146,147,148]
Fucoxanthin	Antioxidant properties Anti-obesity Anti-cancer Anti-malarial effects Bone-protective effects Anti-inflammatory effects Anti-hepatotoxicity effects	N.A.*	[81,96,149,150,151,152,153,154,155,156]
Zeaxanthin	Antioxidant properties Anti-diabetic retinopathy Acute and chronic coronary syndromes prevention Cataract and age-related macular degeneration prevention Visual function maintenance Anti-cardiovascular diseases	As a food coloring agent Used in poultry feeding	[41,134,136,151,157,158,159,160,161,162,163,164,165,166,167,168,169,170,171]
β-Cryptoxanthin	Antioxidant properties Anti-cancer Osteoporosis prevention Bone formation stimulation and protective effects	N.A.*	[151,172,173,174,175,176,177,178]
Bacterioruberin	Antioxidant properties Anti-cancer	N.A.*	[39]
Sioxanthin	Antioxidant properties	N.A.*	[179]
Salinixanthin	Antioxidant properties Anti-cancer	N.A.*	[39]
Saproxanthin	Antioxidant properties Apoptosis-inducing effects	N.A.*	[151,180]
Violaxanthin	Antioxidant properties Anti-inflammatory effects in macrophages	As a food coloring agent	[41,179,181]
Myxol	Antioxidant property Anti-cancer	N.A.*	[151,182]
Echinenone	Antioxidant properties	N.A.*	[81,179,183]
Phytoene	Antioxidant properties Antitumoral activity	N.A.*	[39,144,184]
Siphonaxanthin	Antiproliferative activity	N.A.*	[185]

N.A.*: not available.

**Table 2 marinedrugs-18-00028-t002:** Some carotenoid-producing microalgae.

Name	Carotenoid(s)	Molecular Formula	Production Yield	Source	Reference
*Dunaliella* sp.	β-carotene	C_40_H_56_	100 mg/L	*Dunaliella salina*	[197,198]
	Astaxanthin	C_40_H_52_O_4_	40–45 mg/g	*Dunaliella salina*	[199]
	α-carotene	C_40_H_56_	2.7 ± 0.5 mg/g	*Dunaliella salina*	[200]
	Zeaxanthin	C_40_H_56_O_2_	11.3 ± 1.6 mg/g	*Dunaliella salina*	[200]
	Lutein	C_40_H_56_O_2_	6.6 ± 0.9 mg/g	*Dunaliella salina*	[200]
*Hematococcus* sp.	Astaxanthin	C_40_H_52_O_4_	357 mg/L	*Haematococcus pluvialis*	[201]
*Chlorella* sp.	Astaxanthin	C_40_H_52_O_4_	10.3 mg/L	*Chlorella zofingiensis*	[202]
	Lutein	C_40_H_56_O_2_	153,009.7 μg/g	*Chlorella pyrenoidosa*	[203]
	β-cryptoxantin	C_40_H_56_O	334.9 μg/g	*Chlorella pyrenoidosa*	[203]
	Canthaxanthin	C_40_H_52_O_2_	8.5 mg/g	*Chlorella zofingiensis*	[27]
	Auroxanthin	C_40_H_56_O_4_	38.5 μg/g	*Chlorella pyrenoidosa*	[203]
	Violaxanthin	C_40_H_56_O_4_	38.1 μg/g	*Chlorella pyrenoidosa*	[203]
	Neochrome	C_40_H_56_O_4_	65.2 μg/g	*Chlorella pyrenoidosa*	[203]
	Neoxanthin	C_40_H_56_O_4_	199.7 μg/g	*Chlorella pyrenoidosa*	[203]
	Zeaxanthin	C_40_H_56_O_2_	2170.3 μg/g	*Chlorella pyrenoidosa*	[203]
	β-carotene	C_40_H_56_	4314.3 μg/g	*Chlorella pyrenoidosa*	[203]
	α-carotene	C_40_H_56_	4232.5 μg/g	*Chlorella pyrenoidosa*	[203]
*Scenedesmus* sp.	Lutein	C_40_H_56_O_2_	3.8 mg/L. day	*Scenedesmus almeriensis*	[204,205]
*Spirulina* sp.	β-carotene	C_40_H_56_	296 mg/kg	*Spirulina platensis*	[206,207,208,209]
*Chlorococcum* sp.	Lutein	C_40_H_56_O_2_	1.05 mg/L. h	*Chlorococcum citriforme*	[210]
*Tetraselmis* sp.	Astaxanthin	C_40_H_52_O_4_	2.3 mg/g	*Tetraselmis suecica*	[211]
*Muriellopsis* sp.	Lutein	C_40_H_56_O_2_	35 mg/L	Not mentioned	[194]
*Phaeodactylum* sp.	Fucoxanthin	C_42_H_58_O_6_	15.7 mg/g	*Phaeodactylum tricornutum*	[212]
*Galdieria* sp.	Lutein	C_40_H_56_O_2_	0.6 ± 0.1 mg/g	*Galdieria sulphuraria*	[213]
*Coccomyxa* sp.	Lutein	C_40_H_56_O_2_	5–6 g/L	*Coccomyxa onubensis*	[214]
*Odontella* sp.	Fucoxanthin	C_42_H_58_O_6_	80 mg/L	*Odontella aurita*	[215]
*Isochrysis* sp.	Fucoxanthin	C_42_H_58_O_6_	18.23 ± 0.54 mg/g	*Isochrysis aff. Galbana*	[216]
*Chaetoceros* sp.	Fucoxanthin	C_42_H_58_O_6_	2.24 ± 0.01 mg/g	*Chaetoceros gracilis*	[216]
*Nitzschia* sp.	Fucoxanthin	C_42_H_58_O_6_	4.92 ± 0.11 mg/g	*Nitzschia closterium*	[216]

**Table 3 marinedrugs-18-00028-t003:** Some carotenoid-producing marine bacteria.

Name	Carotenoid(s)	Molecular Formula	Production Yield	Source	Reference
*Gordonia* sp.	Canthaxanthin	C_40_H_52_O_2_	0.73 mg/L	*Gordonia jacobaea MV-1*	[217]
*Micrococcus* sp.	Canthaxanthin	C_40_H_52_O_2_	1.70 mg/L	*Micrococcus roseus*	[218]
*Dietzia* sp.	Canthaxanthin	C_40_H_52_O_2_	5.31 mg/L	*Dietzia natronolimnaea* HS-1	[219]
*Brevibacterium* sp.	Canthaxanthin	C_40_H_52_O_2_	9.3 mg/L	*Brevibacterium* KY-4313	[220]
*Haloferax* sp.	Canthaxanthin	C_40_H_52_O_2_	2,194.09 ± 0.3 μg/L	*Haloferax alexandrinus*	[221]
	β-carotene	C_40_H_56_	189.91 ± 0.5 μg/L	*Haloferax alexandrinus*	[221]
	Bacterioruberin	C_50_H_76_O_4_	3,818.45 ± 0.01 μg/L	*Haloferax alexandrinus*	[221]
	3-Hydroxyechinenone	C_40_H_54_O_2_	250.95 ± 0.9 μg/L	*Haloferax alexandrinus*	[221]
*Agrobacterium* sp.	Astaxanthin	C_40_H_52_O_4_	89.7 μg/L	*Agrobacterium aurantiacum*	[222]
	Adonixanthin	C_40_H_54_O_3_	323.38 μg/L	*Agrobacterium aurantiacum*	[222]
*Halorubrum* sp.	Bacterioruberin	C_50_H_76_O_4_	11.47 mg/L	*Halorubrum* sp. TBZ126	[223]
	Lycopene	C_40_H_56_	0.104 mg/L	*Halorubrum* sp. TBZ126	[223]
	β-carotene	C_40_H_56_	0.128 mg/L	*Halorubrum* sp. TBZ126	[223]
*Gramella* sp.	Zeaxanthin	C_40_H_56_O_2_	N.D.*	*Gramella oceani*	[167,224]
				*Gramella planctonica*	[224,225]
*Aquibacter* sp.	Zeaxanthin	C_40_H_56_O_2_	N.D.*	*Aquibacter zeaxanthinifaciens*	[224,225]
*Kordia* sp.	Zeaxanthin	C_40_H_56_O_2_	N.D.*	*Kordia aquimaris*	[224,225]
*Saprospira* sp.	3*R*-saproxanthin	C_40_H_56_O_2_	N.D.*	*Saprospira grandis*	[226,227]
*Flavobacterium* sp.	3*R*,2′*S*-myxol	C_40_H_56_O_3_	N.D.*	*Flavobacterium* sp. strain P99-3	[226,227]
*Anabaena* sp.	3R,2′S-myxol	C_40_H_56_O_3_	N.D.*	*Anabaena variabilis*	[226,227]
*Halobacterium* sp.	α-bacterioruberin	C_50_H_76_O_4_	N.D.*	*Halobacterium salinarum* strain NRC-1 and strain R1	[39,228]
				*Halobacterium sodomense*	
*Haloarcula* sp.	α-bacterioruberin	C_50_H_76_O_4_	N.D.*	*Haloarcula vallismortis*	[39,228]
*Salinibacter* sp.	Salinixanthin	C_61_H_92_O_9_	N.D.*	*Salinibacter ruber*	[228,229]
*Mesoflavibacter* sp.	Zeaxanthin	C_40_H_56_O_2_	N.D.*	*Mesoflavibacter zeaxanthinifaciens*	[224,230]
*Zeaxanthinibacter* sp.	Zeaxanthin	C_40_H_56_O_2_	N.D.*	*Zeaxanthinibacter* *Enoshimensis*	[224,231]
*Muricauda* sp.	Zeaxanthin	C_40_H_56_O_2_	N.D.*	*Muricauda lutaonensis*	[224,232]
*Siansivirga* sp.	Zeaxanthin	C_40_H_56_O_2_	N.D.*	*Siansivirga zeaxanthinifaciens*	[224,233]

N.D.*: not determined.

**Table 4 marinedrugs-18-00028-t004:** Non-exhaustive recent examples of antioxidant exopolysaccharide (EPS)-producing marine bacteria.

Marine Bacteria	EPS Concentration	DPPH• Scavenging	O_2_^−^• Scavenging	Free Radical Scavenging	Nitric Oxide RadicalScavenging	Ferrous Ions Chelation Capacity	Reduction of Ferric Ions Power ^(1)^	Lipid Peroxidation	•OH Scavenging	Reference
*Aerococcus uriaeequi*	20–100 µg/mL (O_2_^−^•)50–250 µg/mL (•OH)	-	12–85%	-	-	-	-	-	20–55%	[261]
*Alteromonas* sp. PRIM-21	0.25–1.0 mg/mL	IC_50_ = 0.61 mg/mL	IC_50_ = 0.33 mg/mL	-	-	-	7.5–20 µg eq. ascorbic acid	-	-	[262]
*Bacillus amyloliquefaciens* 3MS 2017	100 µg/mL	86%	61%	38%	64%	64%	0.003	60%	62%	[263]
	250 µg/mL	98%	72%	43%	76%	70%	0.005	69%	74%	
	500 µg/mL	99%	83%	47%	85%	70%	0.005	75%	85%	
*Bacillus alvei*	20–100 mg/mL	96%	-	-	-	-	-	-	-	[255]
*Bacillus anthracis*	20–100 mg/mL	82%	-	-	-	-	-	-	-	
*Bacillus brevis*	20–100 mg/mL	77%	-	-	-	-	-	-	-	
*Bacillus circulans*	20–100 mg/mL	98%	-	-	-	-	-	-	-	
*Bacillus insolitus*	20–100 mg/mL	85%	-	-	-	-	-	-	-	
*Bacillus licheniformis*	20–100 mg/mL	97%	-	-	-	-	-	-	-	
*Bacillus licheniformis* UD061	5–250 mg/L	-	43%	-	-	-	0.35	-	51%	[254]
*Bacillus marinus*	20–100 mg/mL	83%	-	-	-	-	-	-	-	[255]
*Bacillus polymyxa*	20–100 mg/mL	84%	-	-	-	-	-	-	-	
*Bacillus* sp. MSHN2016	30–200 μg/mL	EC_50_ = 77 μg/mL	-	-	-	-	-	-	-	[256]
*Bacterium polaribacter* sp. SM1127	(0.1)–10.0 mg/mL	55%	28%	-	-	-	-	-	52%	[259]
*Edwardsiella tarda*	ETW1: 8 mg/mL	88%	-	-	-	-	-	79%	89%	[260]
	ETW2: 8 mg/mL	77%	-	-	-	-	-	71%	77%	
*Enterobacter* sp. PRIM-26	0.25–1.0 mg/mL	IC_50_ = 0.44 mg/mL	IC_50_ = 0.65 mg/mL	-	-	-	5–12.5 µg eq. ascorbic acid	-	-	[262]
*Janibacter* sp. TKB-1	<50 µg/mL	69%	-	-	-	-	-	-	-	[257]
*Microbacterium aurantiacum* FSW-25	0.1–3.5 mg/mL	20–80%	23–90%	-	-	-	0.9–1.7	-	25–90%	[258]
*Nitratireductor* sp. PRIM-24	0.25–1.0 mg/mL	IC_50_ = 0.49 mg/mL	none	-	-	-	10–24 µg eq. ascorbic acid	-	-	[262]
*Pseudomonas* PF-6	0–3 mg/mL	IC_50_ = 180 µg/mL	IC_50_ = 149 µg/mL	-	-	-	0.2–0.8	-	IC_50_ = 340 µg/mL	[264]
*Pseudomonas* sp. KKB-1	<50 µg/mL	63%	-	-	-	-	-	-	-	[257]
*Pseudomonas stuzeri* 273	5–20 µg/mL	-	IC_50_ = 20 μg/mL	-	-	-	-	-	IC_50_ = 60 μg/mL	[253]
*Staphylococcus* sp.	20–100 mg/mL	80%	-	-	-	-	-	-	-	[255]
*Zunongwangia profunda* SM-A87	(0.1)–10 mg/mL	49%	27%	-	-	-	-	-	59%	[265]

DPPH•: 1,1diphenyl-2-picryl-hydrazyl free radical; %: corresponds to inhibition %; ^(1)^ In absorbance (A_700_); ETW1 and ETW2: Two water-soluble extracellular polysaccharides derived from *Edwardsiella tarda*.

**Table 5 marinedrugs-18-00028-t005:** Antioxidant assays performed on EPS from microalgae.

Microalgae	EPS Composition	Antioxidant and Scavenging Activity Determination Assays	Reference
*Navicula* sp.	Glc, Rha, Gal, Man, Xyl, protein, sulfate	DPPH, ABTS radical scavenging assays	[280]
*Pavlova viridis*	N.D.*	DPPH, LPO, hydroxyl and superoxide anion radical scavenging assays, mouse red blood cell hemolysis assay	[281]
*Porphyridium* sp.	Xyl, Glc, Gal, Ara, Rha, Man, GlcA, sulfates	TBA, FOX, and DCFH assays	[24]
*Porphyridium cruentum*	Xyl, Glc, Gal, GlcA, sulfates	LPO assay in mouse liver homogenates, hydroxyl and superoxide anion radical scavenging assays	[96]
*Rhodella reticulata*	Xyl, Glc, Gal, Rha, GlcA, sulfates	FOX and TAOC assays, hydroxyl and superoxide anion radicals scavenging assays	[276,282]
*Sarcinochrysis marina*	N.D.*	DPPH, LPO, hydroxyl and superoxide anion radicals scavenging assays, mouse red blood cell hemolysis assay	[281]

N.D.*: Not determined, TBA: thiobarbituric acid, FOX: ferrous oxidation-xylenol orange, DCFH: dichlorofluorescein, TAOC: total antioxidant capacity, LPO: lipid peroxidation, DPPH: 1,1-diphenyl-2-picrylhydrazyl radical, Glc: glucose, Rha: rhamnose, Gal: galactose, Man: mannan, Xyl: xylose, Ara: arabinose, GlcA: glucoronic acid.

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
