# Peer review of "Marine Bacteria versus Microalgae: Who Is the Best for Biotechnological Production of Bioactive Compounds with Antioxidant Properties and Other Biological Applications?"

_marinedrugs, 2019, doi:10.3390/md18010028_

Round 1

Reviewer 1 Report

The manuscript is often repetitive and it seems that it has not been read well before submission. The title is "bacteria vs microalgae", but after a brief description of bacteria and microalgae, paragraphs include data from both. This makes difficult establishing if one is better than the other (as the title states). Moreover, the manuscript lacks a conclusive paragraph helping the reader about the better producer.

The manuscript contains a number of type-setting errors (please, check). 

It is not clear how cyanobacteria are classified. This creates confusion. If you want to consider cyanobacteria as microalgae (your intention is clear in figure  1 and table 2, for example), you must refer to them as prokaryotic microalgae. Also the general term "microorganisms" should be avoided. In may case, the reader don't immediately understand if you are talking about bacteria or microalgae (and if the term microalgae includes cyanobacteria or not).

Paragraphs 2.1. and 2.2, to avoid confusion, please distinguish better between cyanobacteria (like Spirulina) and eukaryotic microalgae. As for microalgae, I suggest to create a sub-paragraph on cyanobacteria.

Cyanobacteria first appear as bacteria and then as microalgae. In the paragraphs 3.3.2, after the description of eukaryotic microalgae, cyanobacteria appear at line 168. At line 173 the authors distinguish between microalgae and cyanobacteria. 

lines 138-141 are really confusing. All the paragraph 2.2 should be clearer as it is not clear if you refer to eukaryotes or prokaryotes. The paragraph 2.2 should be titled Eukaryotic microalgae maybe. lines 189-192 are again confusing.

l 41, lipids

l 45, organisms

l 168, 194, and paragraph 2.2.1, Chlorella in italics

l 169, microalgae

l. 176, and lines 189-192,  Arthrospira is a cyanobacterium

In my opinion the manuscript needs a general rearrangement and could be accepted after a major revision. 

Author Response

Dear Respected Editors and Reviewers,

Thank you very much for your e-mail forwarding the reviewers’ comments on our manuscript. After receiving the comments, we considered carefully each point raised by the reviewers and revised the manuscript based on the comments. Please kindly find the answers to the comments/suggestions of the reviewers listed seriatim as follows:

Authors: The repetitions have been carefully omitted from the text and the article has been read well before submission. Marine bacteria and microalgae descriptions have been extensively modified and carefully rearranged. To answer the question of which one is better (marine bacteria or microalgae?), a section has been written and placed into the conclusion. A paragraph regarding the better producer has been addressed in the article according to the comments. Type-setting errors have also been carefully taken care of. The classification of cyanobacteria has been rearranged according to the comments. The general term “microorganisms” have also been avoided in order to give the reader clearer definition of the text. In paragraph 2.1 and 2.2, eukaryotic and prokaryotic microalgae have been clearly separated in a way that they do not cause any confusion for the reader. Additional sub-paragraph on cyanobacteria is thought to be unnecessary and causing extra ramification to the text which is more confusing for the readers. Collectively, all comments corresponding to cyanobacteria classification has been addressed. Confusing lines have been heavily modified in order not to cause any confusion for the reader. Paragraph 2.2 has been modified and it is now clear that we are referring to prokaryotic or eukaryotic microalgae. About the title of the paragraph 2.2, sub-paragraphs clearly give messages about eukaryotic or prokaryotic microalgae but still some sub-paragraphs contain general information about the shared features between eukaryotic and prokaryotic microalgae which cannot be separated. Type setting errors and mistakes in lines 41, 45, 168, 169, 176, and 189-192 as well as in paragraph 2.2.1 have all been addressed and taken care of.

Reviewer 2 Report

In this review article the authors aim at inventorying the potential of bacteria and mircoalgae for the production of antioxidant molecules. This topic is of interest for several applications in marine biotechnology and therefore would deserve a review. However I would recommend first the authors to deeply revise the written English but also the organisation of this review. It is really confusing and the messages are not clearly given. Text is too long definitions are missing, focus not clear etc... they should cite the recent review "Promises and Challenges of Microalgal
Antioxidant Production" in the journal Antioxidants from MDPI. They should also explain why they focus only on carotenoids and EPS.. in brief they would need to revise a lot the structure and writing of their document before publication

The introduction on bacteria and microalgae are too long and vague, The authors should better refer as reference papers in this field. They should only focus o antioxidant activities and not cite examples of anticancer etc activities.

Parts 2.1 and 2.2 should therefore be largely reduced and focused on antioxidants activities. For example I do not see the importance of microalgae for wastewater in this particular review. These introductory parts should therefore be largely shortened. Why detail only three groups of microalgae in this part?

Part 3.1 is another long introduction that largely is a repetition of the general introduction of the review.

Table 1 is of interest but the separation between the molecules is not clear

Overall along the reading of this review we are floating between drugs and antioxidant properties. I would recommend the authors to define properly the different antioxidant activities possible and the families of natural products already targeted for their antioxidant activities and the new molecules and species that are promising in this field.

Author Response

Dear Respected Editors and Reviewers,

Thank you very much for your e-mail forwarding the reviewers’ comments on our manuscript. After receiving the comments, we considered carefully each point raised by the reviewers and revised the manuscript based on the comments. Please kindly find the answers to the comments/suggestions of the reviewers listed seriatim as follows:

Authors: The written English has been deeply revised and relative modifications have been applied. Reorganizations have been considered and applied as well. We have tried our best to avoid any misinterpretations and messages have been re-given in an explicit manner. Proper concision has been applied onto the context of the article and focusing have been meticulously bettered. The article “Promises and Challenges of Microalgal Antioxidant Production” in the journal Antioxidants from MDPI has been cited as asked.

Our main focus has been set on carotenoid and exopolysaccharides beside some other bioactive compounds because our own experimental experiences have been in this particular field, so we have proper prospect and notion of the topic.

Since our final title has been changed and focused on antioxidant properties and other biological applications, we tend to slightly focus on various other properties other than antioxidant.

Part 2.1 and 2.2 have been heavily modified and rearranged according to the comments. The application of microalgae in the particular field of wastewater treatment has been discussed in the text. The three mentioned groups of microalgae are the most important ones; therefore, they have been briefly discussed in the text.

Part 3.1 has been extensively modified and repetitions have been put aside.

Table 1 had been arranged according to the journal’s protocols, now they have been rearranged in a different way in order to give a clearer expression of each field (separation between the molecules is easier to comprehend now).

Reviewer 3 Report

This paper needs extensive editing for English language.  There are many typos and grammatical errors.  The subject is interesting but not presented in a interesting manner.  Mostly long lists of organisms and the compounds they produce without any new or interesting insights.  The question in the title is not really answered and there is no conclusion or discussion to wrap up the paper.

Author Response

Dear Respected Editors and Reviewers,

Thank you very much for your e-mail forwarding the reviewers’ comments on our manuscript. After receiving the comments, we considered carefully each point raised by the reviewers and revised the manuscript based on the comments. Please kindly find the answers to the comments/suggestions of the reviewers listed seriatim as follows:

Authors: Extensive English language editing has been carried out on the text and the typos and grammatical errors have all been taken care of according to the comments. The manner of expression and writing has been heavily altered and we hope that they are presented in an interesting manner this time. Interesting insights of the organisms and bioactive compounds have been appended onto the context of the article. An exhaustive conclusion has been added to top the whole paper off, and the question presented in the title has been settled with a clear answer in the conclusion.

Round 2

Reviewer 1 Report

The manuscripts has been modified according my comments. In my opinion, it should be now accepted for publication after a moderate English revision.

Author Response

Dear Respected Editors and Reviewers,

Thank you very much again for your e-mail forwarding the reviewers’ comments on our manuscript.

In order to address the comments made by the editor, we have attached a section to the article that sufficiently discusses “Ovothiols and their properties and potential applications”.

In order to address the comments made by reviewer 1, we asked two of our native English-speaking colleagues to meticulously revise the English of this article. We believe that the article is now free of any type-setting and grammatical errors, and consequently hope for its publication.

Reviewer 2 Report

The authors have addressed most of the comments made and I consider this review could now be published as it is. Congratulations

Author Response

(The authors gave the same response as above.)
